# Palliative Care Nursing in Australia and the Role of the Registered Nurse in Palliative Care

**Rajkumar Cheluvappa** [1,*] and **Selwyn Selvendran** [2]

1   Nursing and Midwifery, Australian Catholic University, Watson, ACT 2602, Australia
2   Department of Surgery, St George Hospital, Kogarah, NSW 2217, Australia
*   Correspondence: rajkumarchel@gmail.com

**Abstract:** The registered nurse has crucial preventative, therapeutic, sociocultural, and advocacy roles in promoting quality holistic patient-centred palliative care. This paper examines, describes, and analyses this multifaceted role from an antipodean perspective. We conducted systematic searches using *PubMed*, *Google Scholar*, government guidelines, authoritative body regulations, quality control guidelines, and government portals pertaining to palliative care nursing in Australia. This paper relies upon the information garnered from publications, reports, and guidelines resulting from these searches and analyses. The fundamental principles and guiding values of palliative care (and nursing) and the raison d'etre for palliative care as a discipline are underscored and expanded on. Australian Clinical Practice Guidelines (CPGs) pertaining to palliative end-of-life (EOL) nursing care and associated services are discussed. The relevant NMBA nursing standards that RNs need to have to administer opioids/narcotics in palliative care are summarised. The identification of patients who need EOL care, holistic person-centred care planning for them, and consultative multidisciplinary palliative clinical decision making are discussed in the palliative care context. Several components of advance care planning apropos health deterioration and conflicts are discussed. Several aspects of EOL care, especially palliative nursing care, are analysed using research evidence, established nursing and palliative care standards, and the Australian EOL CPGs.

**Keywords:** advance care directive; advance care planning; clinical practice guidelines; end-of-life care; law; legislation; medical treatment decision maker; national safety and quality health service standards; national consensus statement; nursing; nursing and midwifery board of Australia; palliative care; palliative care standards; registered nurse

## 1. Palliative Care and Nurse-Led End of Life Care

The World Health Organization (WHO) describes palliative care as "an approach that improves the quality of life of patients (adults and children) and their families who are facing the problems associated with life-threatening illness, through the prevention and relief of suffering by means of early identification and correct assessment and treatment of pain and other problems, whether physical, psychosocial or spiritual" [1–3]. The Australian palliative care authority, Palliative Care Australia (PCA), defines palliative care in the Australian context as, "Palliative care is person and family-centred care provided for a person with an active, progressive, advanced disease, who has little or no prospect of cure and who is expected to die, and for whom the primary treatment goal is to optimise the quality of life" [4]. This PCA emphasises the need for easy access to holistic patient-centred and family-centred multidisciplinary palliative care [4]. Nurse-led end-of-life (EOL) care planning, including advanced care planning, is increasingly acknowledged as a palliative care approach associated with improved patient outcomes, especially in ways that are holistically patient-centred and family-centred [5].

## 2. Research Approach

Information was gathered from publications, reports, and guidelines resulting from *PubMed*, *Google Scholar*, quality control guidelines, and government portals pertaining to palliative care nursing in Australia. The key search words we used included single or multiple combinations of advance care directive, advance care planning, clinical practice guidelines, end of life care, law, legislation, medical treatment decision maker, national safety and quality health service standards, national consensus statement, nursing, nursing and midwifery board of australia, palliative care, palliative care standards, and registered nurse. Publications that defined and promoted quality holistic palliative care nursing were perused for analysis with additional consultation from the Palliative Care Australia (PCA) standards, Nursing and Midwifery Board of Australia (NMBA) standards, National Safety and Quality Health Service (NSQHS) standards, the National Consensus Statement (NCS) on palliative care, and Australian Clinical Practice Guidelines (CPGs) pertaining to palliative EOL nursing care. Papers pertaining to topics of concern in palliative care were perused and referenced. Our meticulous searches resulted in about 20 publications and around a dozen government/official reports. Being a brief report and analytical treatise, this paper does not fall under the purview of the Preferred Reporting Items for Systematic Reviews and Meta-Analyses (PRISMA), as it is not a systematic review or a meta-analysis.

## 3. The Australian National Palliative Care Standards

The Australian National Palliative Care Standards that govern palliative care practice in Australia are compartmentalised into two groups (Figure 1). Standards 1 to 6 list the apparatus and enablers required to give clinical care, and standards 7 to 9 describe the expectations pertaining to quality assurance, assessment, and fine-tuning of services (Figure 1) [4]. The PCA's self assessment tools/resources are used to self-assess conformity to the standards [4].

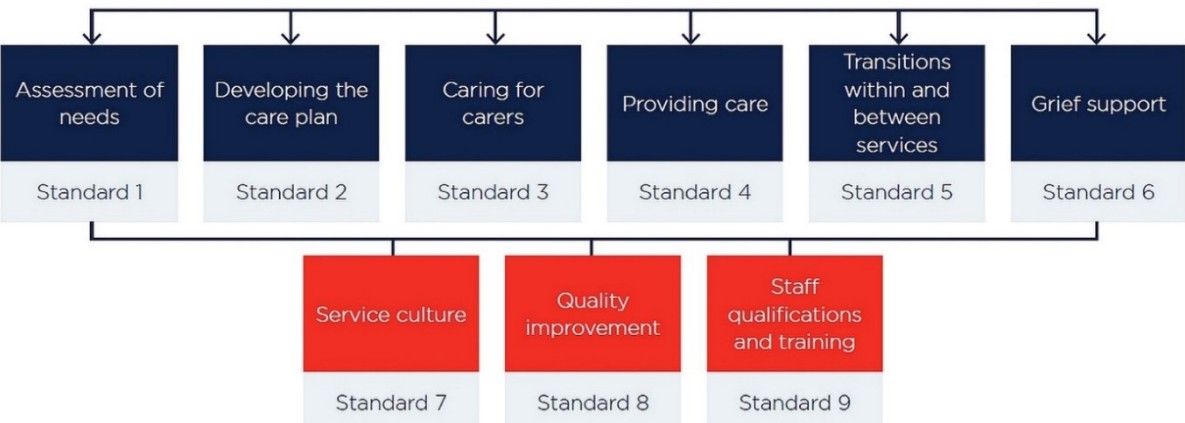

**Figure 1.** The Australian national palliative care standards. Standard 1 pertains to the patient's multi-faceted needs, and standard 2 involves consultative drafting of personalised care plans for patients [4]. Standard 3 addresses the needs of the patient's carers/family, and standard 4 fine-tunes consultative planned care [4]. Standard 5 proffers smooth transitions and interdisciplinary care, and standard 6 tends to grief/loss support for families and carers [4]. Standards 7 and 8 pertain to priming palliative service environment and quality [4]. Standard 9 optimises staff qualifications, efficiency, training, and performance [4].

## 4. The Guiding Values and Principles of Palliative Care

The guiding values of palliative care are dignity, empowerment, compassion, equity, respect, advocacy, excellence, and accountability [4]. Palliative care ameliorates pain and distress and can be incorporated early in the illness trajectory depending on the condition [1,3]. It shows living and dying as a natural process without hastening or

postponing death. Its holistic multi-disciplinary care approach maximises a patient's activity, quality of life, and zest for life owing to the ample inclusion of spiritual and emotional care elements [1,3]. Palliative care also assists in supporting a patient's family.

**5. The National Consensus Statement (NCS) on Palliative Care and the National Safety and Quality Health Service (NSQHS) Standards**

The National Consensus Statement on palliative care summarises the "10 essential elements for safe and high-quality EOL care" [6]. Elements 1 to 5 cover the logistics of planning and delivery of EOL care should [6]. They include "patient-centred communication and shared decision-making, teamwork and coordination of care, components of care, use of triggers to recognise patients approaching the end of life, and response to concerns" [6]. Elements 6 to 10 cover the organisational set-up and establishment prerequisites for delivering good quality, safe EOL care [6]. They include leadership and governance, education and training, supervision and support for interdisciplinary team members, evaluation, audit and feedback, and systems to support high-quality care. The NCS on palliative care corresponds with items listed in the National Safety and Quality Health Service (NSQHS) standards.

**6. The Key Roles of the (Palliative Care) Nurse According to the Nursing and Midwifery Board of Australia (NMBA) Standards and National Safety and Quality Health Service (NSQHS) Standards**

A palliative care RN should have specific attributes, abilities, and training as posited in the NMBA nursing standards [7]. Some of the important ones are standards 1.1/1.4/6.5 (evidence-based nursing and law-/rules-compliant), 1.5 (ethical practice), 1.6 (thorough documentation of interventions and evaluations), 2.2/2.5 (apt communication, patient autonomy, and advocacy), 2.8 (multidisciplinary integrated care and consultative nursing), 2.9/6.6 (prompt reporting of poor clinical practice), 3.4 (accept accountability for decisions), 3.7 (health promotion towards good outcomes), 4.1/4.2 (do holistic assessments), 5 (planned clinical practice), 6.1/6.2 (practice withing scope and safe responsible nursing), and 7 (evaluates condition/changes versus goals, records, communicates, and revises care plan as recommended) [8–10].

The National Safety and Quality Health Service (NSQHS) standards present the standards of care that patients should expect from healthcare institutions. In particular, the "Delivering comprehensive care" segment of the NSQHS standards is relevant, as it outlines strategies and specific actions for providing comprehensive holistic patient-centred, goal-directed, compassionate care to palliative care patients [11]. This NSQHS segment overlaps with NMBA nursing standards 2.2, 2.5, 2.8, 4.1, 4.2, and 7 summarised earlier [8,11].

Commonly used opioids/narcotics in palliative care include morphine, fentanyl, hydromorphone, pethidine, oxycodone, and codeine [12]. A palliative care RN administering opioids to patients should have specific attributes, abilities, and training to administer narcotics [8]. The RN training prerequisites for the use of Schedule 8 drugs (including opioids) are rigorous and enumerated elsewhere [13,14]. The key relevant NMBA nursing standards expected of a RN herein are standards 1.1 (evidence-based practice), 1.4/6.5 (compliance with law and rules), 1.5 (ethics), 1.6 (documentation of assessments and treatments), 2.2 (correct communication), 2.5 (advocates patient's autonomy and legal capacity), 2.8 (collaborative practice), 2.9/6.6 (reporting of malpractices and poor standards), 3.4 (personal and vicarious accountability for actions), 3.7 (promotes better health outcomes), 4.1/4.2 (conducts holistic multipronged assessments), 5 (planning nursing practice), and 6.1/6.2 (providing safe quality care within his/her scope) [8,13]. The palliative care standards to ensure the safe administration of narcotics in palliative care are standard 1 (attending to the patient's multi-faceted needs), standard 3 (caring for patient's carers and family), standard 4 (providing patient-optimised care), and standard 7/8 (ensuring care quality and environment quality) [4].

### 7. The Role of the Registered Nurse in Planning and Preparing for the Management of the Palliative Care Patient Facing Imminent Death

Across the care continuum, the palliative patient's plan of care should be properly developed and constantly updated with changes in his/her condition, circumstances, and family dynamics [15]. Psychosocial, metaphysical, emotional, theological, and cultural considerations should also be incorporated in the care plan and conveyed to the patient in a way he/she understands it well [16]. This will encourage palliative care patients and boost their morale.

Once a patient is identified and assessed at the risk of dying soon, he/she needs to be transitioned to terminal care at the earliest. Whilst the patient is still cognitively competent, it is essential to bring to fruition some certainty regarding his/her death, legacy, and family future. It is in the patient's best interest that his/her preferences, spiritual convictions, and values are explicitly stated, witnessed, and recorded to see that they are adhered to whilst decisions regarding his healthcare are taken if he/she loses the cognitive ability to choose or communicate wishes (decision-making capacity).

This can be done by advance care planning (ACP), which includes appointing an official medical treatment decision maker (MTDM) and/or setting out an advanced care directive (ACD) document [7]. The ACD comprises binding instructions and/or preferences on topics such as treatment venues, final rites, life-prolonging treatments (like intubation or ventilator support), and cardiopulmonary resuscitation [17]. Legislation permits an official MTDM to be appointed to a cognitively competent patient in the presence of two witnesses if the patient's cognition is expected to be lost subsequently [9]. An MTDM can make treatment decisions for the patient if he/she loses the capacity to make them on the person's behalf if they do not have capacity to make the decision [18]. A will/testament can be written up with legal consultation, and an enduring power of attorney (EPOA) can be appointed to make decisions for the patient after loss of mental capacity [15].

Treatment limitation decisions are sometimes difficult to make, especially if the clinicopathological condition is complex and if the opinions of family members or treating clinicians conflict with or between each other [19]. Specific therapy should be withheld from the patient if its risks/complications outweigh the possible benefits, if the patient is projected not to survive it, if it will impair the existing quality of life, if the therapeutic targets could be easily missed, and/or if it is rejected by the patient, his/her MTDM, or his/her ACD document [16,18,19].

The ACP process is crucial for the provision of safe, good-quality EOL care. The NCS on palliative care underscores the pivotal role of the ACP process in providing palliative patients the opportunity to transmit their ethics, objectives, legacies, etc., towards the end of their lives [6]. The NSQHS Standards corresponding to the NCS on palliative are NSQHS Standards 1.18.4, 9.8.1, and 9.8.2 [20,21]. These NSQHS standards include advance care planning (including treatment limitation/cessation orders) in standard 1, Governance for Safety and Quality in Health Service, and standard 9, Recognising and Responding to Clinical Deterioration in Acute Health Care. NSQHS Standards 1.18.4, 9.8.1, and 9.8.2 mandate organisational systems to support the ACP process, ACD crafting, and issuing of treatment limiting/cessation orders towards ensuring the patient's treatment preferences are readily available at any step of the healthcare process/system [20,21].

The palliative care nurse will be part of the multidisciplinary team involved in palliative care planning, which should include regular updating of the patient's goals, aspirations, realistic interests, and family relationship dynamics and, when relevant, reconciling with estranged people [16]. When meticulously considered, these items can/may inject a sense of fulfillment or completion in the patient's life and ease the process of dying [15].

With the palliative care team and patients work together to make care decision, the patient is far more likely to receive treatment compatible with his/her goals of care. This "partnering of care" is commensurate with both the NCS on palliative care and the NSQHS Standards Action 1.18.1 and underscores the importance of shared decision making and patient-centred holistic EOL care [6,20]. It is obvious that patient's and/or MTDM's

wishes should be evidenced and documented officially. The NSQHS Standards Action 1.18.2 requires palliative health services to have mechanisms to support these [20]. Towards this, the NSQHS Standards Action 1.18.3 requires palliative care organisations to have the set up to align the mental capacity of a patient to the information provided to the patient via open communication modalities, comprehensible formats for information dissemination, and communication training to palliative care nurses and clinicians [20].

Conflict minimisation, especially between a patient's family members, is paramount to ensure a terminal patient's feeling of completion/fulfillment. Empathetic but forthright/ clear communication with the patient and his/her family members regarding prognosis, intervention prospects, second opinions, preparing for death, ACP, ACD, MTDM, medicolegal matters, organ donation, ethical practice, and settling family matters will soothe many sore points and minimise conflicts [15]. If a conflict regarding treatment cessation or care approaches cannot be resolved amicably, the chief medical officer may need to review clinical decisions, the evidence for such decisions, communications/correspondences, legal injunctions, and appropriate documentation [18].

## 8. Recognising, Assessing, and Managing Deteriorating Palliative Patients

The transition of a patient to terminal care requires the prompt and perspicacious identification of his/her holistic condition, including the magnitude of recent functional decline, deglutition issues, altered sensorium, impaired intake of nutrition, etc. The NSQHS standards Action 5.14 expands on using comprehensive care plans to monitor efficacy of actions, review, reflect, update, reassess efficacy, and institute care plan changes if the patient's condition changes. This is commensurate with NMBA standards 1.6, 2.8, 3.7, 4.1, 4.2, and 7 [8,9,22]. The NSQHS Standards Action 5.14 and NMBA Standard 7 are to be consistently demonstrated by palliative care nurses via evaluating, recording, investigating, revising care plans, and managing a palliative patient's deterioration [8–10,22]. Factors such as dehydration, infection, polypharmacy, iatrogenic issues, endocrine issues, deranged blood tests, constipation exacerbation, psychiatric manifestations, and renal/cardiovascular issues must be ruled out before a thorough multidisciplinary team assessment is done. The multidisciplinary team, including the palliative care nurse, must use an individualised holistic assessment to determine if further treatment aimed to prolong life is still appropriate or beneficial. The team should consider if the patient has an advance care directive (ACD) to refuse therapy or his/her medical treatment decision maker (MTDM) has refused further treatment.

## 9. Navigating End-of-Life Nursing Care with Multi-Team Involvement

Nursing care pertaining to quality EOL care should adhere to the Australian national palliative care standards [4]. Standard 1 caters to assisting with his multipronged needs, and standard 2 involves including the palliative care patient's wishes in the tailor-made drafting of his/her care plan [4]. Standard 3 involves including catering to the needs of the patient' kin and friends, and standard 4 includes holistic multidisciplinary patient-centred care planning [4]. Standard 5 necessitates easy transitions between care specialties, and standard 6 underscores the support required for grief suffered by the patient's family [4]. Standards 7 and 8 are very important because they hone and optimise the quality of services [4]. Standard 9 relates to the qualifications, work quality, and training requirements of staff [4].

Differences of opinion between a terminal patient's family members and friends are common. Shared decision making regarding treatment continuance, changes, or cessation is made smoother and less traumatic by proactive, prompt, clear, and honest but empathetic communication [10]. However, it must be stressed that the patient's wishes or ACD are paramount and surpass everyone else's [18,23]. In circumstances where an MTDM refuses recommended therapy, but the patient's preferences are unknown, and no ACD is available, the palliative care team must notify the public advocate for review, analysis, and final decision [18]. If a terminal patient or his/her MTDM requests a treatment that is ineffectual, contraindicated, unrealistic, inappropriate, or unnecessary, the palliative care team need

to be candid but kind in communicating why such an intervention is unwarranted [10]. Similar communication is also necessary if the patient refuses necessary treatment [19]. Second opinions and mental health consultations may be arranged to assess and reassure the patient in these circumstances [10,24]. A few selected senior members of the palliative care team are preferably and consistently involved in these discussions with the patient to ensure authority, clarity, and consistency in information transmission and charting out actions acceptable to the patient [10,24].

## 10. Palliative Care Health Professionals

Most health-care professionals are trained to promote and maintain life and often have difficulty when faced with the often-rapid decline and death of people with terminal illnesses. By contrast, data suggest that early and open discussion of EOL issues with patients and families allows time for reflection and planning, can obviate the introduction of unwanted interventions or procedures, can provide reassurance, and can alleviate fear [25,26]. These benefit both patients and palliative health professionals [26,27]. Patients' perspectives regarding EOL interventions and use of technologies might differ from those of the health professionals involved in their care, and health-care professionals should recognise this and respect the patient's autonomy [28]. Advance care directives can preserve autonomy, but their legal validity and use varies between countries. Clinical management of the end of life should aim to maximise quality of life of both the patient and caregiver and, when possible, incorporate appropriate palliation of distressing physical, psychosocial, and existential distress [25,27]. Training of health-care professionals should include the development of communication skills that help to sensitively manage the inevitability of death [29]. The emotional burden for health-care professionals caring for people with terminal disease should be recognised, with structures and procedures developed to address compassion, fatigue, and the moral and ethical challenges related to providing EOL care [27,30].

## 11. Voluntary Assisted Dying in Australian States and Territories

It is out of scope of this paper to discuss Victoria's and Western Australia's new voluntary assisted dying laws and parallel legislation, which are to become active in New South Wales (mid-2024), Tasmania (October 2022), South Australia (December 2022), and Queensland (January 2023) [31,32]. Voluntary assisted dying is currently illegal in the Australian Territories [31,32].

## 12. Conclusions

This paper has substantial well-collated information to offer pertaining to the practice of quality palliative nursing care. The identification hallmarks of patients needing to undergo EOL palliative care were initially discussed. Advance care planning for the palliative patient is essential. The palliative care team should be multidisciplinary, properly trained, and adherent to NMBA, NSQHS, and palliative care standards with the nurse focussing on patient empowerment, dignity, equity, respect, and advocacy. It is better for just one or two senior palliative team clinicians to consistently liaise with the patient and his/her family regarding EOL care and clinical decision making. This paper pinpoints pertinent items of concern and important areas of focus in the palliative care milieu, which are also important topics for teaching and research therein. Palliative care planning should cater to the palliation of psychological, physical, spiritual, and existential distress and should include aspects of the patient's family dynamics and the preclusion of conflicts. The palliative team should consider if the patient has an ACD to autonomously refuse specific treatment therapy or has appointed an MTDM to take decisions on his/her behalf. Forthright and early discussion of EOL care (using good communication skills) with patients and their families permits better quality of the patient's final stages of life relative to the chronological prolongation of life. Such discussions promote contemplation, reassurance, lessening of fear, fulfilling final family commitments, and proper planning to face the inevitable without facing inefficacious clinical interventions.

**Author Contributions:** Conceptualization, R.C.; methodology, R.C.; software, R.C.; validation, R.C.; formal analysis, R.C.; investigation, R.C.; resources, R.C.; data curation, R.C.; writing—original draft preparation, R.C.; writing—review and editing, R.C. and S.S.; visualization, R.C.; supervision, R.C.; project administration, R.C.; funding acquisition, R.C. All authors have read and agreed to the published version of the manuscript.

**Funding:** This research received no external funding.

**Institutional Review Board Statement:** Not applicable.

**Informed Consent Statement:** Not applicable.

**Conflicts of Interest:** The authors declare no conflict of interest.

## Abbreviations

AIHW: Australian Institute of Health and Welfare; ACD: Advance Care Directive; ACP: Advanced Care Planning; ADL: Activities of Daily Living; CPGs: Clinical Practice Guidelines; EOL care: End-of-Life care; MTDM: Medical Treatment Decision Maker; NCS: National Consensus Statement; NMBA: Nursing and Midwifery Board of Australia; PCS: Palliative Care Standards; RN: Registered Nurse; WHO: World Health Organization.

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
