# Peer review of "Palliative Care Nursing in Australia and the Role of the Registered Nurse in Palliative Care"

_nursrep, doi:10.3390/nursrep12030058_

Round 1
Reviewer 1 Report
This paper is not a research paper, but as a review report, I agree with the author's intention to inform non-Australian readers on palliative end-of-life (EOL) and related literature based on Australian nurses' perspectives on palliative care. I think that the revised manuscript is more refined than the original manuscript which was somewhat distracting.
Author Response
This paper is not a research paper, but as a review report, I agree with the author's intention to inform non-Australian readers on palliative end-of-life (EOL) and related literature based on Australian nurses' perspectives on palliative care.
- Thank you
I think that the revised manuscript is more refined than the original manuscript which was somewhat distracting.
- Thank you
Reviewer 2 Report
A review article that provides a broad perspective on the role of the registered nurse in aged care and palliative care in Australia. The article reviews key legal and other documents on the topics under consideration. It will provide an important summary and guide for nursing care practice, particularly in Australia. It has been improved by the amendments that have been suggested. I also present some suggestions to make it more improved:
(a) the key words are not in line with the article, it presents key word from the previous article, it has not eliminated the elder's words;
b) I think that the objective of this short report should be revised: that only examine, or also describe and analyze the role of nurses in this context?
c) I think that at the point of the research approach, they could be more rigorous and describe that the search for key words in the databases found x articles of relevance to this report (all focusing on Australia). Although they state that they will not use PRISMA, this data would be important for methodological rigor, even though it is a theoretical article.
d) I would also suggest that the conclusions, despite being a brief report, should be more comprehensive and state the consequences of this brief report for the practice of care, for teaching and for research itself.
Author Response
A review article that provides a broad perspective on the role of the registered nurse in aged care and palliative care in Australia. The article reviews key legal and other documents on the topics under consideration. It will provide an important summary and guide for nursing care practice, particularly in Australia. It has been improved by the amendments that have been suggested. I also present some suggestions to make it more improved:
- Thank you.
(a) the key words are not in line with the article, it presents key word from the previous article, it has not eliminated the elder's words;
- All irrelevant keywords have been removed in the amended manuscript, and several new relevant keywords have been added.
(b) I think that the objective of this short report should be revised: that only examine, or also describe and analyze the role of nurses in this context?
Towards this,
- The title of the paper has been changed to “Palliative care nursing in Australia and the role of the registered nurse in palliative care”
- The sentence (in the Abstract), “This paper examines this role from an antipodean perspective” has been replaced by “This paper examines, describes, and analyses this multifaceted role from an antipodean perspective”.
(c) I think that at the point of the research approach, they could be more rigorous and describe that the search for key words in the databases found x articles of relevance to this report (all focusing on Australia). Although they state that they will not use PRISMA, this data would be important for methodological rigor, even though it is a theoretical article.
Towards this, the following 3 sentences have been added to the Research approach segment:
- “The key search words we used included single or multiple combinations of Advance Care Directive, Advance Care Planning, Clinical Practice Guidelines, End of Life Care, Law, Legislation, Medical Treatment Decision Maker, National Safety and Quality Health Service Standards, National Consensus Statement, Nursing, Palliative Care, Palliative Care Standards, and Registered Nurse”
- “Our meticulous searches resulted in about 20 publications and around a dozen government/official reports”
- “Being a brief report and analytical treatise, this paper does not fall under the purview of the Preferred Reporting Items for Systematic Reviews and Meta-Analyses (PRISMA), as it is not a systematic review or a meta-analysis”
(d) I would also suggest that the conclusions, despite being a brief report, should be more comprehensive and state the consequences of this brief report for the practice of care, for teaching and for research itself.
Towards this, the following sentences have been placed in suitable places:
- This paper has substantial well-collated information to offer pertaining to the practice of quality palliative nursing care.
- This paper pinpoints pertinent items of concern and important areas of focus in the palliative care milieu, which are also important topics for teaching and research therein.
This manuscript is a resubmission of an earlier submission. The following is a list of the peer review reports and author responses from that submission.
Round 1
Reviewer 1 Report
This paper is not a research paper, but rather addresses Australia's role of the registered nurse in aged and palliative care in the Code of Conduct from the Nursing and Midwifery Board of Australia (NMBA), Current Australian law and recent federal legislation, and Australian Clinical Practice Guidelines (CPGs). through pertaining to palliative end of life (EOL) and related literature.
As suggested in the title of this study, this study is based on perspectives relating to Australia or New Zealand, and it is difficult to see this paper as a paper considering international readers.
Author Response
Dear Editor/Reviewer,
In accordance with Reviewer 2’s comments, we have split the manuscript into 2 individual manuscripts. We have systematically responded to your comments/notes below and amended our manuscripts accordingly (uploaded). Please note that our amendments cannot be made in the MDPI “Manuscript for Revisions” as they include extensive EndNote citation changes and reordering. We have used our original submission manuscript to make amendments. Please consider our 2 new revised manuscripts positively for publication.
As specified earlier, we have also split the manuscript into 2 individual manuscripts. The segment on funding has been completely removed. Several sub-headings have been removed and multiple topics compacted into fewer ones. The 2 manuscripts have been streamlined with smoother flow between topics and sub-topics.
Sincerely yours,
Rajkumar Cheluvappa
=========================
Reviewer 1
This paper is not a research paper, but rather addresses Australia's role of the registered nurse in aged and palliative care in the Code of Conduct from the Nursing and Midwifery Board of Australia (NMBA), Current Australian law and recent federal legislation, and Australian Clinical Practice Guidelines (CPGs). through pertaining to palliative end of life (EOL) and related literature.
Thank you for your comments. Our manuscript has been submitted as a review paper, not an original research paper.
=========================
As suggested in the title of this study, this study is based on perspectives relating to Australia or New Zealand, and it is difficult to see this paper as a paper considering international readers.
Thank you for your comments. Our manuscript is essential reading to an international audience outside of ANZ owing to the counter-perspectives it offers, undergirded by guidelines and legislation, towards reflecting on and improving one’s own aged care nursing perspectives.
Reviewer 2 Report
A review article that gives a broad perspective on the role of the registered nurse in aged care and palliative care in Australia. The article revisits key legal and other documents on the topics under consideration. It will provide an important summary and guide for nursing care practice, particularly in Australia. As the article does not make connections between aged care and palliative care (presenting conclusions independent of the parts) I suggest that instead of one article you write two independent articles. Unless you can justify this option of one article, it does not seem scientifically coherent. In the summary of the article itself, we have two summaries and I believe that this is not the correct way to present the work, except if there is a reasoned justification and to connect the topics under study
I would also like to be more elucidated about the point "The role of the palliative care nurse in planning and preparing for the management of the palliative care patients facing imminent death" on the role of nurses in the promotion/elaboration of advance care planning. Are nurses the ones who make or promote the discussion on advance care planning?
Also in the section "Palliative care health professionals", the lack of preparation of professionals to deal with the end of life is mentioned. They suggest that health professionals are too focused on treatment and little on end-of-life care and that there is a need to discuss this issue with the person and the family. These statements are good but need bibliographic support, that is, there is a need to point out some references on death education and cite them at this point (this aspect was not seen in the references).
Author Response
Dear Editor/Reviewer,
In accordance with Reviewer 2’s comments, we have split the manuscript into 2 individual manuscripts. We have systematically responded to your comments/notes below and amended our manuscripts accordingly (uploaded). Please note that our amendments cannot be made in the MDPI “Manuscript for Revisions” as they include extensive EndNote citation changes and reordering. We have used our original submission manuscript to make amendments. Please consider our 2 new revised manuscripts positively for publication.
As specified earlier, we have also split the manuscript into 2 individual manuscripts. The segment on funding has been completely removed. Several sub-headings have been removed and multiple topics compacted into fewer ones. The 2 manuscripts have been streamlined with smoother flow between topics and sub-topics.
Sincerely yours,
Rajkumar Cheluvappa
Selwyn Selvendran
=========================
Reviewer 2
A review article that gives a broad perspective on the role of the registered nurse in aged care and palliative care in Australia. The article revisits key legal and other documents on the topics under consideration. It will provide an important summary and guide for nursing care practice, particularly in Australia.
- Thank you very much.
As the article does not make connections between aged care and palliative care (presenting conclusions independent of the parts) I suggest that instead of one article you write two independent articles. Unless you can justify this option of one article, it does not seem scientifically coherent. In the summary of the article itself, we have two summaries and I believe that this is not the correct way to present the work, except if there is a reasoned justification and to connect the topics under study
- We have split the manuscript into 2 individual manuscripts
- The segment on funding has been completely removed
- Several sub-headings have been removed and multiple topics compacted into fewer ones
- The 2 manuscripts have been streamlined with smoother flow between topics and sub-topics
=========================
I would also like to be more elucidated about the point "The role of the palliative care nurse in planning and preparing for the management of the palliative care patients facing imminent death" on the role of nurses in the promotion/elaboration of advance care planning. Are nurses the ones who make or promote the discussion on advance care planning?
- Towards clarifying this, sentence fragments have been introduced in different parts of the text leading up to advance care planning, including:
- (Nurse-led End of Life (EOL) care planning)”, including advanced care planning,” is increasingly…
- “The palliative care nurse will be part of the multidisciplinary team involved in palliative care planning which…”
- (The multidisciplinary team), “including the palliative care nurse,”(must use an individualised holistic…)
=========================
Also in the section "Palliative care health professionals", the lack of preparation of professionals to deal with the end of life is mentioned. They suggest that health professionals are too focused on treatment and little on end-of-life care and that there is a need to discuss this issue with the person and the family. These statements are good but need bibliographic support, that is, there is a need to point out some references on death education and cite them at this point (this aspect was not seen in the references).
- Thank you for pointing this out to us. Towards this, we have added around 6 or 7 new references (citations in text) in this segment now.
=========================